# Integrated Metabolomic and Transcriptomic Analysis Reveals Potential Gut-Liver Crosstalks in the Lipogenesis of Chicken

**DOI:** 10.3390/ani13101659

**Published:** 2023-05-17

**Authors:** Can Chen, Weilin Chen, Hao Ding, Genxi Zhang, Kaizhou Xie, Tao Zhang

**Affiliations:** 1College of Animal Science and Technology, Yangzhou University, Yangzhou 225009, China; chencan19981224@163.com (C.C.); a934263381@163.com (W.C.); gxzhang@yzu.edu.cn (G.Z.); kzxie@yzu.edu.cn (K.X.); 2Joint International Research Laboratory of Agriculture and Agri-Product Safety, Ministry of Education, Yangzhou University, Yangzhou 225009, China; 3College of Veterinary Medicine, Yangzhou University, Yangzhou 225009, China; 15138343214@163.com

**Keywords:** gut–liver axis, chicken, HFD-induced obese model, lipogenesis, metabolomics, transcriptomics

## Abstract

**Simple Summary:**

The gut microbiota can regulate lipid metabolism with its metabolic products through the gut–liver axis. In the present study, using an HFD-induced obese chicken model, we performed a multiple omics analysis using metabolomics and transcriptomics to identify gut–liver crosstalks involved in regulating the lipogenesis of chicken. The results showed that 5-hydroxyisourate, alpha-linolenic acid, bovinic acid, linoleic acid, and trans-2-octenoic acid might serve as signal molecules between the gut and liver. In the liver, they might enhance the expression of *ACSS2*, *PCSK9*, and *CYP2C18* and down-regulate one or more genes of *CDS1*, *ST8SIA6*, *LOC415787*, *MOGAT1*, *PLIN1*, *LOC423719*, and *EDN2* to promote the lipogenesis of chicken. Moreover, taurocholic acid might be transported from the gut to the liver and contribute to HFD-induced lipogenesis by regulating the expression of ACACA, FASN, AACS, and LPL in the liver. This study lays the foundations for further elucidation of the gut–liver crosstalk mechanisms underlying lipogenesis in chickens.

**Abstract:**

Growing evidence has shown the involvement of the gut–liver axis in lipogenesis and fat deposition. However, how the gut crosstalk with the liver and the potential role of gut–liver crosstalk in the lipogenesis of chicken remains largely unknown. In this study, to identify gut–liver crosstalks involved in regulating the lipogenesis of chicken, we first established an HFD-induced obese chicken model. Using this model, we detected the changes in the metabolic profiles of the cecum and liver in response to the HFD-induced excessive lipogenesis using ultra-high-performance liquid chromatography–tandem mass spectrometry (UHPLC-MS/MS) analysis. The changes in the gene expression profiles of the liver were examined by RNA sequencing. The potential gut–liver crosstalks were identified by the correlation analysis of key metabolites and genes. The results showed that a total of 113 and 73 differentially abundant metabolites (DAMs) between NFD and HFD groups were identified in the chicken cecum and liver, respectively. Eleven DAMs overlayed between the two comparisons, in which ten DAMs showed consistent abundance trends in the cecum and liver after HFD feeding, suggesting their potential as signaling molecules between the gut and liver. RNA sequencing identified 271 differentially expressed genes (DEGs) in the liver of chickens fed with NFD vs. HFD. Thirty-five DEGs were involved in the lipid metabolic process, which might be candidate genes regulating the lipogenesis of chicken. Correlation analysis indicated that 5-hydroxyisourate, alpha-linolenic acid, bovinic acid, linoleic acid, and trans-2-octenoic acid might be transported from gut to liver, and thereby up-regulate the expression of *ACSS2*, *PCSK9*, and *CYP2C18* and down-regulate one or more genes of *CDS1*, *ST8SIA6*, *LOC415787*, *MOGAT1*, *PLIN1*, *LOC423719*, and *EDN2* in the liver to enhance the lipogenesis of chicken. Moreover, taurocholic acid might be transported from the gut to the liver and contribute to HFD-induced lipogenesis by regulating the expression of *ACACA*, *FASN*, *AACS*, and *LPL* in the liver. Our findings contribute to a better understanding of gut–liver crosstalks and their potential roles in regulating chicken lipogenesis.

## 1. Introduction

In recent decades, the growth rate and meat production performance of chickens have been greatly improved, benefiting from advancements in genetic selection, breeding, and feed science. However, it is unfortunate that the deposition of fat, particularly abdominal fat, increases significantly with increasing growth rate, impairing feed efficiency, meat production, reproduction performance, and disease resistance ability of chicken. Therefore, it remains challenging to reduce fat deposition while maintaining a high growth rate in chickens [1,2,3]. Chicken fat deposition is a complex biological process featuring lipogenesis, adipogenesis, and lipid metabolism. Understanding the molecular mechanisms underlying fat deposition is crucial for controlling excessive fat deposition in chickens, contributing to the improvement of meat production and feed efficiency. Adipose tissue is the major site of lipogenesis in mammals, while it is liver tissue in chickens. In chickens, up to 90% of lipid biosynthesis happens in the liver, indicating that elucidating the mechanism of hepatic lipogenesis is critical to developing strategies to control the excessive fat deposition of chickens [4,5].

Recently, the gut–liver axis has gained increasing attention due to its significant effect on liver functions. Through the biliary tract, systemic circulation, and portal vein, the gut and liver are able to communicate with one another. Through the portal vein, metabolites generated by the gut microbiota can reach the liver and affect how it functions [6]. Growing evidence shows that the gut microbiota plays a critical role in regulating lipogenesis and fat deposition by its metabolic products through the gut–liver axis [7,8,9]. However, the role of the gut–liver axis in lipogenesis and fat deposition of chicken remains poorly understood.

In recent years, the multi-omics analysis approach, including metabolomics and transcriptomics, has been widely applied to investigate the role of the gut–liver axis in regulating liver function. For example, Saeedi et al. [10] used an integrated metabolomics and transcriptomics analysis to explore the mechanisms of oxidative liver injury, finding that the gut microbiota-produced metabolite 5-methoxyindoleacetic acid could travel from the gut to the liver and activate nuclear factor erythroid 2-related factor 2 to increase the hepatic resistance to oxidative injury. Using a combined analysis of transcriptome and metabolome, Yang et al. [11] proved that red yeast rice prevents obesity and lowers blood lipids in rats by regulating lipid metabolism pathways in the liver. Mayneris-Perxachs et al. [12] performed a multi-omics study that revealed significant crosstalk among the gut, iron status, and hepatic lipid accumulation. Mechanically, the gut microbiota affects iron status and hepatic fat accumulation by regulating gene expression and signaling pathways in the liver.

In this study, to explore the crosstalks between the gut and liver and investigate the potential roles of the gut–liver axis in regulating lipogenesis and fat deposition in chickens, we first induced excessive lipogenesis and fat deposition in chickens using a high-fat diet (HFD). Then, we examined and compared the cecal and hepatic metabolomics of chickens fed with a normal-fat diet (NFD) and HFD to identify key metabolites related to the enhanced lipogenesis through the gut–liver axis. Finally, we characterized the changes in gene expression profiles after HFD treatment and analyzed their correlation with the key metabolites, aiming to investigate the possible mechanisms that the gut–liver axis uses to regulate lipogenesis in chickens. The results could contribute to a better understanding of the involvement of the gut–liver axis in chicken lipogenesis.

## 2. Materials and Methods

### 2.1. Animals

All animal experiments were performed following the approval of the Animal Welfare Committee of Yangzhou University (permit number SYXK [Su] 2016-0020). Twenty 4-week-old chickens were selected and randomly divided into the NFD group (*n* = 10) and HFD group (*n* = 10). No significant difference in the mean body weight was observed between the two groups (*p* > 0.05). Chickens were housed in separate cages with water and food ad libitum. Chickens in the NFD group were fed with a basic feed. Chickens in the HFD group were fed with basal feed + soybean oil (20%) for four weeks to induce excessive lipogenesis and fat deposition.

### 2.2. Sample Preparation

After being fed with NFD and HFD for four weeks, five chickens were randomly selected from each group. The blood was collected via the wing vein. The serum was collected by centrifugation at 4000 rpm for 5 min. All chickens were anesthetized with an intravenous injection of pentobarbital sodium (50 mg/kg BW) and slaughtered. The body weight and abdominal fat weight were recorded. The liver tissues and cecal content were collected and stored at −80 °C after liquid nitrogen freezing. 

### 2.3. Detection of Lipogenic Indicators

According to the manufacturer’s instructions, serum triglyceride (TG), total cholesterol (TC), low-density lipoprotein (LDL), and high-density lipoprotein (HDL) were measured using the TG, TC, LDL-C, and HDL-C assay kits (Jiancheng, Nanjing, China). As directed by the manufacturer’s instructions, a TG kit (Solarbio, Beijing, China) was used to measure the hepatic TG. Student’s *t*-test was used to detect differences in the lipogenic indicators between NFD and HFD groups.

### 2.4. Metabolomics Analysis

In total, 50 mg of sample (cecal content and liver) was weighed and transformed into a centrifuge tube. The extract solvent (acetonitrile–methanol–water, 2:2:1), which contains an internal standard, was then added in a volume of 1000 L. In the ice-water bath, the sample was vortexed for 30 s and sonicated for 5 min, three times. The sonicated sample was centrifuged at 12,000 rpm for 15 min at 4 °C after being allowed to stand at −20 °C for 2 h. The supernatants were transformed into a sample injection bottle and subjected to ultra-high-performance liquid chromatography–tandem mass spectrometry (UHPLC-MS/MS) analysis. UHPLC-MS/MS analysis was performed using a 1290 UHPLC system (Agilent) coupled with an Orbitrap Q Exactive Focus mass spectrometer (Thermo Fisher Scientific, Waltham, MA, USA).

Using the R package XCMS, base-line filtering, peak identification, integration, retention time correction, peak alignment, and normalization were applied to the MS raw data obtained from the UHPLC-MS/MS analysis in order to produce a peak matrix that contained the retention time (RT), mass-to-charge ratio (*m*/*z*) values, and peak intensity. The peaks were annotated using OSI-SMMS 1.0 software based on an in-house MS/MS database BiotreeDB (V2.1). Metabolites were identified qualitatively based on the first-order MS (MS1) and second-order MS (MS2) data. A metabolite was considered successfully identified if MS1 |ppm| < 10 and MS2 score > 0.6.

The partial least squares discriminant analysis (PLS-DA) was performed using SIMCA-P 14.1 software. The goodness and validity of the fitted PLS-DA model were assessed by a permutation test (200 iterations). The variable importance of project (VIP) value was calculated to evaluate the importance of a metabolite. Metabolites with VIP > 1 and fold change (FC) > 1.5 were considered differentially abundant metabolites (DAMs). Pathway enrichment of metabolites was performed using the online tool Metaboanalyst 5.0 (http://www.metaboanalyst.ca/, accessed on 4 April 2023). A pathway with a *p*-value < 0.05 was considered significantly enriched.

### 2.5. Hepatic Transcriptome Analysis

According to the manufacturer’s instructions, total RNA was isolated from liver tissues using the Trizol reagent kit (Invitrogen, Carlsbad, CA, USA). RNase-free agarose gel electrophoresis and an Agilent 2100 Bioanalyzer (Agilent Technologies, Santa Clara, CA, USA) were used to evaluate the quality of the RNA. The rRNAs were removed to obtain mRNAs. The enriched mRNAs were fragmented into short fragments using a fragmentation buffer and reversely transcribed into the first cDNA with random primers. Second-strand cDNA was synthesized and purified with a QiaQuick PCR extraction kit (Qiagen, Venlo, The Netherlands). The purified cDNA was end-repaired, poly(A)-added, and ligated to Illumina sequencing adapters. Then, the second-strand cDNA was digested using Uracil-N-Glycosylase. The digested products were size selected by agarose gel electrophoresis, PCR amplified, and sequenced using Illumina HiSeqTM 4000 by Gene Denovo Biotechnology Co. (Guangzhou, China).

The raw data from the sequencing were subjected to quality control using the fastp software. Clean readings were produced by eliminating adapter-containing, low-quality reads with more than 50% of bases with a Q-value of 20 and more than 10% of unknown nucleotides (N). The rRNA sequences were filtered by mapping the clean reads to the rRNA database of chicken using bowtie2. Then, the retained reads were mapped to the chicken (*Gallus gallus*) genome GRCg6a (Ensembl release 106) using HISAT2 to identify genes. The gene expression was calculated using the fragments per kilobase of transcript per million mapped reads (FPKM) method. The principal component analysis (PCA) was performed with R package gmodels. Differential gene expression analysis between the NFD and HFD groups was analyzed using the DESeq2 1.22.2 software. Differentially expressed genes (DEGs) were identified with *p*-value < 0.05 and FC > 2. The expression profiles of the DEGs were visualized using the R package pheatmap. Gene Ontology (GO) and Kyoto Encyclopedia of Genes and Genomes (KEGG) enrichment analysis was performed using the R package ClusterProfiler 4.0. GO terms or KEGG pathways with a Q-value < 0.05 were considered significantly enriched.

### 2.6. Correlation Analysis

The Pearson correlations between metabolites and DEGs were calculated using the R package psych. A *p*-value value ≤ 0.05 was considered statistically significant. The correlation between the metabolites and DEGs was visualized by heatmap. The correlation heatmap was plotted by R package pheatmap.

## 3. Results

### 3.1. HFD Induced Enhanced Lipogenesis and Excessive Fat Deposition

We first induced enhanced lipogenesis and fat deposition in chickens using an HFD. As shown in Figure 1, chickens fed with an HFD showed significantly increased body weight (*p* < 0.01), abdominal fat weight (*p* < 0.01), serum TG (*p* < 0.05), serum HDL (*p* < 0.01), serum LDL (*p* < 0.05), and hepatic TG (*p* < 0.01). The results indicated that the chicken obese model with enhanced lipogenesis and excessive fat deposition was established successfully.

### 3.2. Effects of HFD on Cecal Metabolomics

To examine the effect of an HFD on gut metabolic profiles, we analyzed the cecal metabolomics of chickens fed with an NFD and an HFD using LC-MS/MS-based untargeted metabolomics. A total of 17,972 metabolic peaks were detected, containing 9377 in positive mode and 8595 in negative mode. Then, 2097 metabolites were identified qualitatively, including 1333 in positive mode and 764 in negative mode (Appendix A). We constructed a PLS-DA model using the SIMCA-P 14.1 software to identify DAMs between the cecal content of chickens fed with NFD vs. HFD. As shown in Figure 2A,B, the PLS-DA model demonstrates excellent goodness of fit and can effectively distinguish samples between the two groups. Using the PLS-DA model, 113 DAMs were identified, consisting of 67 up-regulated metabolites and 45 down-regulated metabolites (Figure 2C, Appendix A).

Then, we subjected the DAMs to pathway enrichment analysis to analyze metabolic changes in the cecum associated with the HFD-induced lipogenesis and fat deposition. The results showed that six significantly enriched metabolic pathways were identified, including phenylalanine, tyrosine and tryptophan biosynthesis, phenylalanine metabolism, aminoacyl-tRNA biosynthesis, purine metabolism, valine, leucine and isoleucine biosynthesis, and ubiquinone and other terpenoid-quinone biosynthesis (Figure 2D).

### 3.3. Effects of HFD on Hepatic Metabolomics

Then, we analyzed the effects of HFD-induced obesity on hepatic metabolomics by detecting the changes in hepatic metabolite profiles using LC-MS/MS-based untargeted metabolomics. A total of 16,330 metabolic peaks were detected, containing 8779 in positive mode and 7551 in negative mode. Using the thresholds of MS1 |ppm| < 10 and MS2 score > 0.6, 2118 metabolites were identified qualitatively, including 1,432 metabolites in positive mode and 686 metabolites in negative mode (Appendix A). Then, we constructed a PLS-DA model using the SIMCA-P 14.1 software to identify DAMs between liver samples of chickens fed with NFD vs. HFD. As shown in Figure 3A,B, the PLS-DA model indicates high goodness of fit and can effectively distinguish samples between the two groups. Using the PLS-DA model, 73 DAMs were identified, containing 38 up-regulated and 35 down-regulated metabolites (Figure 3C, Appendix A).

We then performed a KEGG pathway enrichment analysis of these DAMs to explore the metabolic changes associated with HFD-induced lipogenesis. As present in Figure 3D, the DAMs were assigned to four significant enriched KEGG pathways, including alpha-linolenic acid metabolism, glyoxylate and dicarboxylate metabolism, linoleic acid metabolism, and the biosynthesis of unsaturated fatty acids.

### 3.4. Effects of HFD on Hepatic Transcriptome

To identify genes involved in HFD-induced lipogenesis, we examined the hepatic transcriptomes of chickens fed with an NFD and an HFD using RNA-seq. A total of 16,845 genes were detected from ten liver samples of chicken. The PCA results showed that HFD treatment alters the mRNA expression profile in the liver (Figure 4A). We compared the gene expression profiles of chicken fed with NFD vs. HFD and identified 271 DEGs between the two groups (Figure 4B,C, Appendix A). The HFD feeding up-regulated 162 genes and down-regulated 109 genes. GO enrichment analysis of the DEGs identified 19 significant enriched biological process GO terms, including many lipid metabolism-related terms such as sterol metabolic process, cholesterol metabolic process, sterol biosynthetic process, lipid biosynthetic process, lipid metabolic process, steroid metabolic process, steroid biosynthetic process, and cholesterol biosynthetic process (Figure 4D, Table 1). Moreover, the DEGs were assigned to nine significantly enriched molecular function terms such as oxidoreductase activity, monooxygenase activity, and heme binding, as well as five cellular component terms such as extracellular space, extracellular region part, and extracellular region (Appendix A).

KEGG pathway enrichment analysis identified steroid biosynthesis, steroid hormone biosynthesis, and retinol metabolism as significantly enriched pathways (Figure 4E).

### 3.5. Identification of Potential Lipogenesis-Related Gut–Liver Crosstalks

To identify potential gut–liver crosstalks, we first selected DAMs both in the cecum and liver with the same abundance trend between NFD and HFD chickens and then analyzed their correlation with the expression of DEGs. Figure 5A showed that 11 DAMs overlayed in the cecum and liver, in which 11 metabolites showed a consistent abundance trend (Table 2, Figure 5B). The abundance of 2-Oxo-4-methylthiobutanoic acid, 4-Hydroxycinnamic acid, and ethyl oleate decreased both in the cecum and liver of chickens fed with an HFD compared with chickens fed with an NFD. In contrast, significantly increased abundances of 5-hydroxyisourate, alpha-linolenic acid, bovinic acid, linoleic acid, lorazepam glucuronide, taurocholic acid, and trans-2-octenoic acid were observed in the cecum and liver of chickens fed with an HFD. The above results suggested that these ten metabolites might be involved in the gut–liver crosstalk. Functional enrichment analysis indicates that the ten common metabolites were assigned to seven pathways, including four significant enriched pathways: the biosynthesis of unsaturated fatty acids, linoleic acid metabolism, taurine and hypotaurine metabolism, and alpha-linolenic acid metabolism (Figure 5C).

The GO annotation (Table 1) indicated that 35 DEGs were involved in the lipid metabolic process. Figure 5D displayed the expression profiles of the 35 DEGs across the NFD and HFD samples. Then, we selected these 35 DEGs as potential genes regulated by the gut and analyzed their correlations with the ten common metabolites in the liver to identify metabolite–gene interactions between the gut and liver. The results demonstrated two interesting clusters. Cluster 1 contains three genes, including *ACSS2* (Acyl-CoA synthetase short-chain family member 2), *PCSK9* (proprotein convertases ubtilisin/kexin type 9), and *CYP2C18* (cytochrome P450 family 2 subfamily C member 18), which were positively correlated with the abundance of 5-hydroxyisourate, alpha-linolenic acid, bovinic acid, linoleic acid, and trans-2-octenoic acid in the liver. A negative correlation was observed between the expression of these three genes and the abundance of ethyl oleate. Another cluster includes seven genes such as *CDS1* (CDP-diacylglycerol synthase 1), *ST8SIA6* (ST8 alpha-N-acetyl-neuraminide alpha-2,8-sialyltransferase 6), *LOC415787*, *MOGAT1* (monoacylglycerol O-acyltransferase 1), *PLIN1* (perilipin 1), *LOC423719*, and *EDN2* (endothelin 2). Most of these seven genes showed positive correlations with the abundance of ethyl oleate and negative correlations with the abundance of 5-hydroxyisourate, alpha-linolenic acid, bovinic acid, linoleic acid, and trans-2-octenoic acid in the liver. In addition, we also found that some important lipogenesis-related genes, including *ACACA* (acetyl-CoA carboxylase alpha), *FASN* (fatty acid synthase), *AACS* (acetoacetyl-CoA synthetase), and *LPL* (lipoprotein lipase), were positively correlated with the abundance of taurocholic acid (Figure 5E).

## 4. Discussion

Recently, the gut–liver axis has attracted growing attention for its importance in controlling gastrointestinal health and disease [13]. Numerous studies have also revealed the critical role of the gut–liver axis in regulating lipogenesis and fat deposition in mammals [7,8,9]. Our earlier research showed that chicken lipogenesis is facilitated by gut microbiota dysbiosis via modifying metabolomics in the cecum [14], indicating the potentially critical role of the gut in regulating chicken lipogenesis. However, it is uncertain whether the gut crosstalk with the liver regulates the lipogenesis and fat deposition of chicken. The HFD-induced obese animal model has been widely used to reveal the causal relationship between gut microbiota and the liver in the development of obesity [15,16,17]. In the present study, we constructed an HFD-induced obesity model to study the crosstalks between the gut and liver associated with the lipogenesis of chicken. HFD treatment for 60 days significantly promoted chicken lipogenesis and fat deposition, indicating the successful establishment of the obese chicken model.

Then, we detected the changes in cecal and hepatic metabolic profiles following HFD treatment using LC-MS/MS-based metabolomics technology. In total, 113 and 73 DAMs were identified in the cecum (NFD vs. HFD) and liver (NFD vs. HFD), respectively. Eleven DAMs overlayed between the two comparisons, in which ten metabolites showed a consistent abundance trend in the cecum and liver, including 2-oxo-4-methylthiobutanoic acid, 4-hydroxycinnamic acid, 5-hydroxyisourate, alpha-linolenic acid, bovinic acid, ethyl oleate, linoleic acid, lorazepam glucuronide, taurocholic acid, and trans-2-octenoic acid, suggesting that they might serve as signal molecules between the gut and liver. Functional enrichment analysis showed that these ten DAMs were significantly enriched into three lipid metabolism pathways. The above results imply the potential of the ten metabolites as signaling molecules in the gut–liver crosstalk associated with the HFD-induced lipogenesis of chicken. 

Through the portal vein, the microbiota in the gut can deliver metabolites to the liver that affect liver function by influencing hepatic gene expression [13,18]. Therefore, we identified DEGs between liver samples of chicken in the NFD and HFD groups and analyzed their correlations with the ten important metabolites. The results showed that HFD treatment significantly up-regulated 162 and down-regulated 109 genes in the liver, respectively. GO and KEGG enrichment analysis showed that DEGs were enriched into lipid metabolism-related biological processes and pathways, suggesting that the HFD feeding significantly altered the expression profiles of lipid metabolism-related genes and influenced the lipogenesis of chicken. Interestingly, we found that 35 DEGs were enriched into lipid metabolism-related biological processes. We hypothesized that the gut-produced metabolites might be transported from the gut to the liver and then regulate the expression of these 35 DEGs to affect hepatic lipogenesis and fat deposition in chickens. Thus, we analyzed the correlations between the 35 DEGs and these ten potential signal metabolites. We found that three up-regulated genes, *ACSS2*, *PCSK9*, and *CYP2C18*, were significantly positively correlated with 5-hydroxyisourate, alpha-linolenic acid, bovinic acid, linoleic acid, and trans-2-octenoic acid and negatively correlated with ethyl oleate.

As an acyl-CoA synthetase short-chain family member, *ACSS2* is essential for lipogenesis because it produces acetyl-CoA from acetate [19]. By selectively regulating lipid metabolism-related genes, *ACSS2* enhances mice’s systemic fat accumulation and utilization [20]. In our study, *ACSS2* was up-regulated in chickens fed with an HFD and showed the most significant fold change, indicating its critical role in the lipogenesis of chickens. *PCSK9* plays an important role in cholesterol homeostasis and intracellular lipogenesis [21]. In C57BL/6 J mice, HFD can promote de novo hepatic *PCSK9* expression and elevate circulating PCSK9 levels, contributing to liver fat accumulation [22]. The increased *PCSK9* significantly promotes the hepatic apoB-containing TRL production/secretion, accompanied by hepatic lipid accumulation, as well as increased plasma levels of VLDL-triglycerides, triacylglycerol, and apoB [23,24,25]. In the avian species, *CYP2C18* is a crucial subfamily of cytochrome P450 and a factor in susceptibility to chemical substances having xenobiotic metabolic activity [26]. In chicken, *CYP2C18* is highly enriched in liver tissue, indicating its potential role in regulating hepatic lipogenesis [27]. Taken together, the above studies indicated that *ACSS2*, *PCSK9*, and *CYP2C18* play key roles in lipogenesis, and the up-regulated 5-hydroxyisourate, alpha-linolenic acid, bovinic acid, linoleic acid, and trans-2-octenoic acid might serve as gut–liver signals to promote HFD-induced lipogenesis and fat deposition of chicken by enhancing the expression of *ACSS2*, *PCSK9*, and *CYP2C18* in the liver. 

In addition, we also found that seven genes, including *CDS1*, *ST8SIA6*, *LOC415787*, *MOGAT1*, *PLIN1*, *LOC423719*, and *EDN2*, were negatively correlated with the abundance of 5-hydroxyisourate, alpha-linolenic acid, bovinic acid, linoleic acid, and trans-2-octenoic acid. *CDS1* encodes CDP-diacylglycerol synthase 1 and catalyzes the conversion of phosphatidic acid into CDP-DAG. In Hela cells and 3T3-L1 preadipocytes, when CDS1 was knocked down, lipid droplets grew to massive or supersized sizes, and 3T3-L1 preadipocyte development was largely inhibited [28,29], showing an anti-lipogenic function of *CDS1*. *MOGAT1* is a de novo lipogenesis gene that converts monoacylglycerol to diacylglycerol. In mice, *MOGAT1* may play a role in adipocyte differentiation in vitro [30]. On an HFD, global Mogat1 deletion contributed to obesity, insulin sensitivity, and glucose intolerance [31]. *PLIN1* regulates the access of lipases to neutral lipids in lipid drops, making it one of the most prevalent lipid droplet-related proteins on the surface of adipocytes and playing a critical role in lipid homeostasis [32]. To date, the roles of *ST8SIA6*, *LOC415787*, *LOC423719*, and *EDN2* in lipid metabolism are rarely reported. As concluded by the above results, the increased abundance of 5-hydroxyisourate, alpha-linolenic acid, bovinic acid, linoleic acid, and trans-2-octenoic acid might inhibit one or more genes of *CDS1*, *ST8SIA6*, *LOC415787*, *MOGAT1*, *PLIN1*, *LOC423719*, and *EDN2* in the liver, enhancing lipogenesis and fat deposition. 

*ACACA* encodes the acetyl-coenzyme A carboxylase 1 (ACC1), the primary and rate-limiting enzyme in the de novo biosynthesis of fatty acid [33]. ACC1 catalyzes the conversion of acetyl-CoA into de novo lipogenesis substrate malonyl-CoA. In livestock, the *ACACA* gene has been confirmed to be a potential candidate gene for fatness and milk fat traits [34,35]. As a key enzyme, FASN is involved in long-chain fatty acid de novo synthesis. In the presence of NADPH, FASN catalyzes the decarboxylation condensation of acetyl CoA, malonyl CoA, and other small carbon units to produce palmitate [36]. *AACS* is an enzyme that uses ketone bodies to convert acetoacetate into acetoacetyl-CoA, which then gives acetyl units as the precursors for lipogenesis [37]. The knockdown of *AACS* inhibits the differentiation of 3T3-L1 cells. In chicken, plasma *AACS* showed significant positive associations with hepatic lipid deposition and can serve as a biomarker for non-alcoholic fatty liver disease [38]. The LPL gene encodes lipoprotein lipase, the master regulator of fatty acid uptake from triglyceride-rich lipoproteins. In mice fed with an HFD, the knockdown of the LPL gene reduces adiposity and improves plasma insulin and adipokines [39]. Tibet kefir milk decreases fat deposition by regulating the gut microbiota and gene expression of LPL in high-fat-diet-fed rats [40]. In the present study, the abundance of taurocholic acid significantly increased after HFD feeding and showed a positive correlation with the expression of *ACACA*, *FASN*, *AACS*, and *LPL*, implying that taurocholic acid might be transported from the gut to the liver and contribute to the HFD-induced lipogenesis and fat deposition by regulating the expression of *ACACA*, *FASN*, *AACS*, and *LPL* in the liver. 

## 5. Conclusions

In summary, this study established an HFD-induced obese chicken model to investigate the potential crosstalks between the gut and liver associated with the lipogenesis of chicken. We found that HFD feeding affected the metabolic profiles in the cecum and liver and altered gene expression profiles in the liver. Correlation analysis indicated that 5-hydroxyisourate, alpha-linolenic acid, bovinic acid, linoleic acid, and trans-2-octenoic acid might be transported from gut to liver, and thereby up-regulate the expression of *ACSS2*, *PCSK9*, and *CYP2C18* and down-regulate one or more genes of *CDS1*, *ST8SIA6*, *LOC415787*, *MOGAT1*, *PLIN1*, *LOC423719*, and *EDN2* in the liver to enhance the lipogenesis of chicken. Moreover, taurocholic acid might be transported from the gut to the liver and contribute to HFD-induced lipogenesis by regulating the expression of *ACACA*, *FASN*, *AACS*, and *LPL* in the liver. Our findings contribute to a better understanding of gut–liver crosstalks and their potential roles in regulating chicken lipogenesis.

## Figures and Tables

**Figure 1 animals-13-01659-f001:**
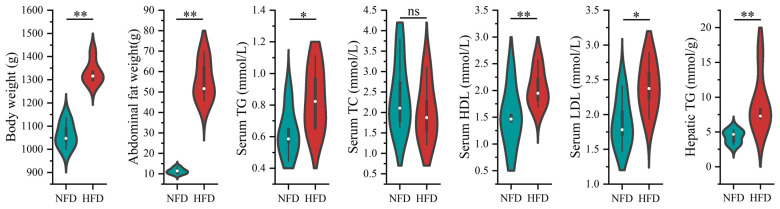
The effect of HFD on lipogenesis of chickens. The green and red colored violins represent normal-fat diet (NFD) and high-fat diet groups, respectively. The difference in the lipogenic indicators between NFD and HFD groups was compared using the Student’s *t*-test. * indicates *p* < 0.05, ** indicates *p* < 0.01, and ns indicates *p*  ≥ 0.05.

**Figure 2 animals-13-01659-f002:**
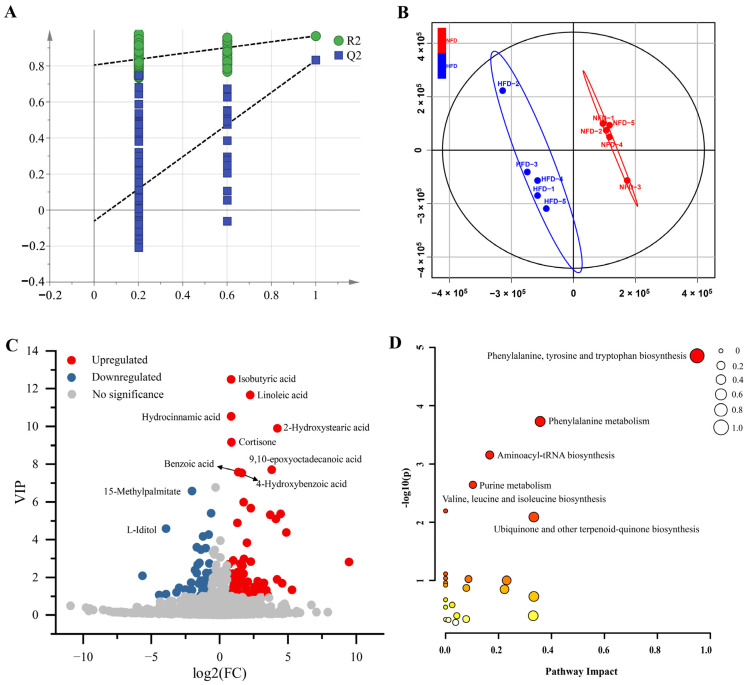
DAMs between the cecum of chickens fed with NFD and HFD. (**A**) Permutation test of the PLS-DA model (200 times). The blue Q2 regression line crosses the vertical axis (on the left) below zero, and all of the blue Q2 values to the left are lower than the original points to the right. These features strongly suggest that the PLS-DA model that was created is valid. (**B**) Scores scatter plot of the samples based on the PLS-DA model. (**C**) Volcano plot of the DAMs. (**D**) The bubble plot of the KEGG pathway enrichment analysis. The size of the bubble represents the pathway impact. The color ranging from orange to red indicates *p*-value from high to low.

**Figure 3 animals-13-01659-f003:**
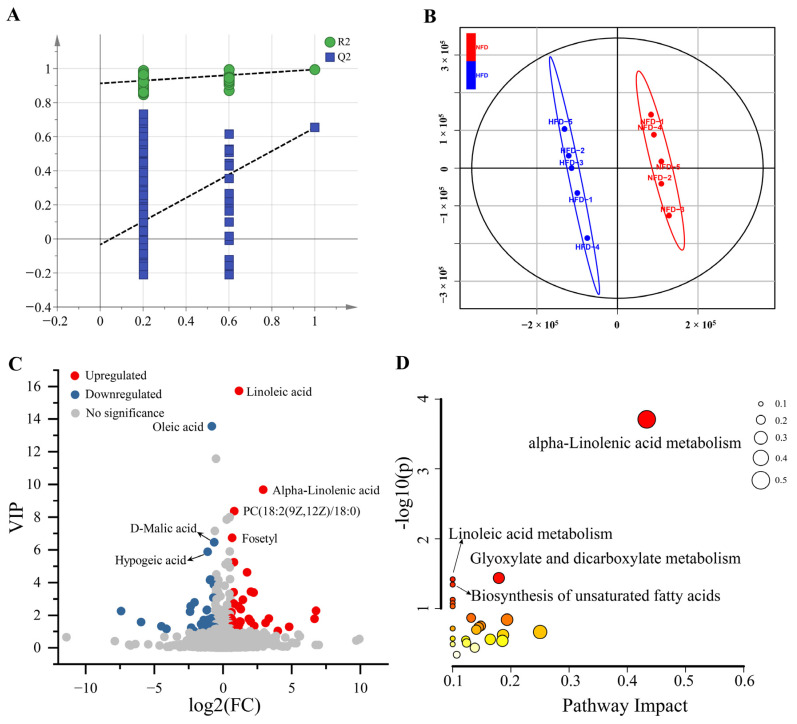
DAMs between the liver of chickens fed with NFD and HFD. (**A**) Permutation test of the PLS-DA model (200 times). The blue regression line of the Q2 points intersects the vertical axis (on the left) below zero, strongly indicating that the constructed PLS-DA model is valid. (**B**) Scores scatter plot of the samples based on the PLS-DA model. (**C**) Volcano plot of the DAMs. (**D**) The bubble plot of the KEGG pathway enrichment analysis. The size of the bubble represents the pathway impact. The color ranging from orange to red indicates *p*-value from high to low.

**Figure 4 animals-13-01659-f004:**
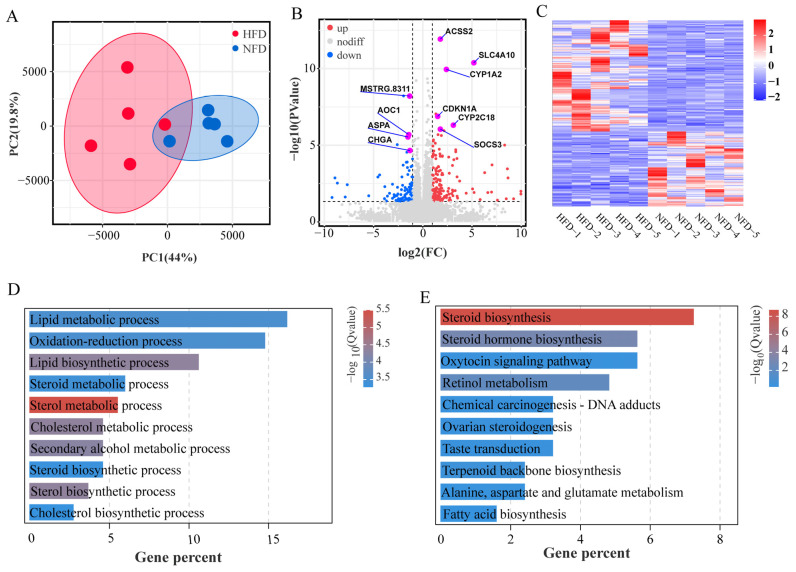
Effect of HFD on the hepatic transcriptome. (**A**) PCA score plot of the liver samples from two groups based on all expressed genes. (**B**) Volcano plot of differentially expressed genes. (**C**) Heatmap of differential gene expression. (**D**,**E**) Top ten significant enriched GO biological processes and KEGG pathways.

**Figure 5 animals-13-01659-f005:**
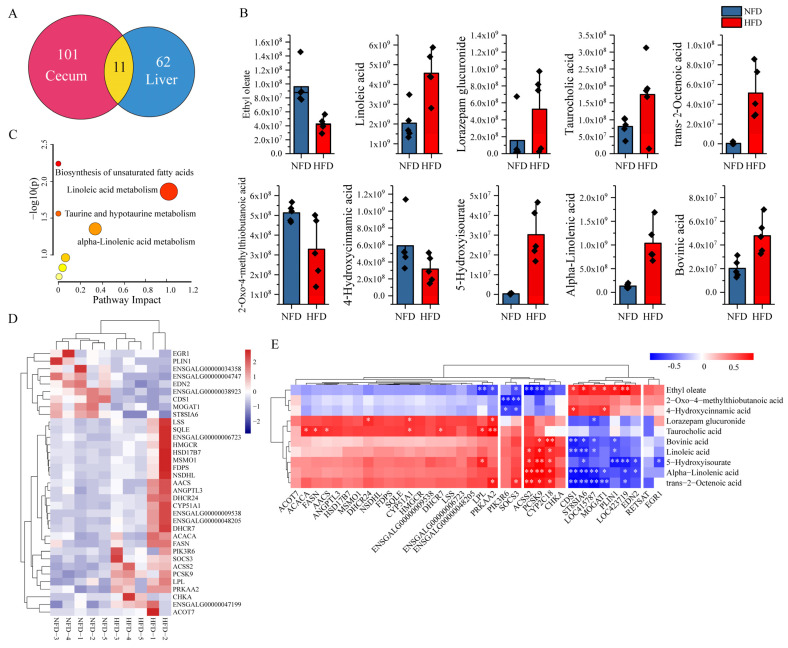
Correlation analysis of key metabolites and genes. (**A**) Venn plot of the DAMs in the cecum and liver. Eleven DAMs overlayed between the cecum and liver. (**B**) Visualization of the abundance of ten DAMs with a consistent trend in the cecum and liver. (**C**) Pathway enrichment analysis of the ten DAMs with a consistent trend in cecum and liver. (**D**) Expression heatmap of the 35 DEGs involved in lipid metabolic process. (**E**) Correlation between the ten DAMs and the 35 DEGs. * indicates *p* < 0.05, ** indicates *p* < 0.01.

**Table 1 animals-13-01659-t001:** The significantly enriched GO biological processes terms.

GO ID	Description	Gene Number	Q Value
GO:0016125	sterol metabolic process	12	3.14 × 10^−6^
GO:0008203	cholesterol metabolic process	10	4.35 × 10^−5^
GO:0016126	sterol biosynthetic process	8	4.35 × 10^−5^
GO:0008610	lipid biosynthetic process	23	4.47 × 10^−5^
GO:1902652	secondary alcohol metabolic process	10	5.40 × 10^−5^
GO:0006629	lipid metabolic process	35	2.98 × 10^−4^
GO:0055114	oxidation-reduction process	32	2.98 × 10^−4^
GO:0008202	steroid metabolic process	13	3.85 × 10^−4^
GO:0006694	steroid biosynthetic process	10	4.77 × 10^−4^
GO:0006695	cholesterol biosynthetic process	6	5.24 × 10^−4^
GO:1902653	secondary alcohol biosynthetic process	6	1.04 × 10^−3^
GO:0006066	alcohol metabolic process	15	1.67 × 10^−3^
GO:0044283	small molecule biosynthetic process	19	2.37 × 10^−3^
GO:1901615	organic hydroxy compound metabolic process	17	6.11 × 10^−3^
GO:0044711	single-organism biosynthetic process	36	6.11 × 10^−3^
GO:0042742	defense response to bacterium	9	9.74 × 10^−3^
GO:0046165	alcohol biosynthetic process	9	2.36 × 10^−2^
GO:0034128	negative regulation of MyD88-independent Toll-like receptor signaling pathway	2	4.62 × 10^−2^
GO:0071615	oxidative deethylation	2	4.62 × 10^−2^

**Table 2 animals-13-01659-t002:** Differential metabolites with common trend between cecum and liver in response to HFD treatment.

Metabolites	Cecum		Liver	
	Fold Change	VIP	Fold Change	VIP
2-oxo-4-methylthiobutanoic acid	5.632986445 ↓	1.34903	1.562114 ↓	3.84847
4-hydroxycinnamic acid	4.022521132 ↓	1.31592	1.874103 ↓	4.17537
5-hydroxyisourate	9.492578339 ↑	1.68904	101.3498 ↑	1.77908
alpha-linolenic acid	22.38871776 ↑	5.35261	7.669327 ↑	9.67412
bovinic acid	7.363542529 ↑	1.26022	2.359367 ↑	1.61691
ethyl oleate	1.581726323 ↓	1.64936	2.253551 ↓	2.29809
linoleic acid	4.777977301 ↑	11.6529	2.239847 ↑	15.7242
lorazepam glucuronide	13.1473488 ↑	5.31738	3.366851 ↑	4.61575
taurocholic acid	10.84230807 ↑	1.31562	2.177994 ↑	2.53774
trans-2-octenoic acid	718.8764339 ↑	2.81436	109.9878 ↑	2.26858

Note: ↓ indicates down-regulated abundance of a metabolite in chickens fed with HFD compared with NFD. ↑ indicates up-regulated abundance of a metabolite in chickens fed with HFD compared with NFD.

## Data Availability

The data presented in this study are available on request from the corresponding author. The data are not publicly available due to privacy.

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
