# Peer review of "Integrated Metabolomic and Transcriptomic Analysis Reveals Potential Gut-Liver Crosstalks in the Lipogenesis of Chicken"

_animals, 2023, doi:10.3390/ani13101659_

Round 1

Reviewer 1 Report

Dear authors,

the proposed scientific manuscript is scientifically based and well presented (this is my opinion).
I have only four minor suggestions:
1) I suggest that instead of (p < 0.05) you state the actual (real) p-values (line 167 and 168). If you accept the suggestion, the text in line 107 "A p-value < 0.05 indicates a significant difference." is not necessary.
2) line 131; instead of "hepatic ..." write "Hepatic ..."
3) line 259; instead of "... alpha-Linolenic acid ..." write "... alpha-linolenic acid ..."

4) line 391; instead of "conclusion" write "Conclusion"

Author Response

Dear reviewer,

We really appreciate your helpful comments and valuable suggestions. The comments have been addressed and corrected. We hope that the present manuscript now is suitable for publication in Animals. All the corrections and modifications are as follows:

Reviewer #1

  1. I suggest that instead of (p < 0.05) you state the actual (real) p-values (line 167 and 168). If you accept the suggestion, the text in line 107 "A p-value < 0.05 indicates a significant difference." is not necessary.

Response: We accepted this suggestion and deleted the text "A p-value < 0.05 indicates a significant difference." in line 107.

  1. line 131; instead of "hepatic ..." write "Hepatic ..."

Response: The "hepatic" has been changed to "Hepatic".

  1. line 259; instead of "... alpha-Linolenic acid ..." write "... alpha-linolenic acid ..."

Response: The "alpha-Linolenic acid" has been changed to "alpha-linolenic acid". Furthermore, we also checked this issue throughout the manuscript and corrected this misspelling.

  1. line 391; instead of "conclusion" write "Conclusion"

Response: The "conclusion" has changed to "Conclusion".

Thanks again for your suggestions!

Reviewer 2 Report

In this study, authors used an HFD-induced obese chicken model to perform a multiple omics analysis using metabolomics and transcriptomics to identify gut-liver crosstalks involved in regulating the lipogenesis of chicken. In my opinion, although this paper has some obvious shortcomings (e.g., lack of qPCR validation), this manuscript has some scientific value, but some parts need to be revised and improved.

1.     Suggest adding the HFD-induced obese model to the keywords.

2.     The authors should add progress in the application of metabolomics and transcriptomics in relevant studies in the INTRODUCTION section.

3.     Line 87, twenty should be Twenty.

4.     Authors should indicate in the methods the database for metabolite identification.

5.     Line 176, omit the word “method”.

6.     Line 177, “Using the thresholds of MS1 |ppm| <10 and MS2 177 score >0.6” this part should be moved to method section.

7.     Figure 3D, the number of metabolites represented by each bubble should be marked on the figure.

8.     Figure 4D, GO annotations are generally grouped into three main categories: Biological Process (BP), Cellular Component (CC) and Molecular Function (MF). Authors should show comprehensive GO enrichment results, or at least present them in the supplementary material.

9.     They carried metabolomics of cecal and hepatic, but only transcriptome of hepatic. Why? They performed metabolomics of the cecal and hepatic, but only the transcriptome of the hepatic. Please clarify.

10.   Same question. In section 3.5, the authors selected DAMs both in the cecum and liver with the same abundance trend between NFD and HFD chickens and then analyzed their correlation with the expression of DEGs, while the DEGs are based on the liver transcriptome. Could the author please explain it?

11.   Line 307, “Ten DAMs overlayed between the two comparisons.” But in Figure 5A, it is eleven.

12.   Line 398, omit the word “genes”.

Although there are some minor errors, this article is well-written in English.

Author Response

Dear reviewer,

We really appreciate your helpful comments and valuable suggestions. The comments have been addressed and corrected. We hope that the present manuscript now is suitable for publication in Animals. All the corrections and modifications are as follows:

Reviewer #2

  1. Suggest adding the HFD-induced obese model to the keywords.

Response: The “HFD-induced obese model” has been added to the Keywords.

  1. The authors should add progress in the application of metabolomics and transcriptomics in relevant studies in the INTRODUCTION section.

Response: The progress in the application of metabolomics and transcriptomics in the gut-liver axis study has been added to the Introduction section (line 77-89 ).

  1. Line 87, twenty should be Twenty.

Response: The twenty has been corrected to Twenty.

  1. Authors should indicate in the methods the database for metabolite identification.

Response: Our study used an in-house MS/MS database named BiotreeDB (V2.1). This information has been added to the manuscript (line150-151).

  1. Line 176, omit the word "method".

Response: The word "method" has been deleted (line 219).

  1. Line 177, "Using the thresholds of MS1 |ppm| <10 and MS2 177 score >0.6" this part should be moved to method section.

Response: This part has been moved to the 2.4 section (line 153-154).

  1. Figure 3D, the number of metabolites represented by each bubble should be marked on the figure.

Response: We are so sorry that we made a mistake in describing Figure 2D and 3D. We checked the raw data of the pathway enrichment analysis using MetaboAnalyst 5.0. Indeed, the node size represents the pathway impact. We have corrected this mistake throughout the manuscript. And the legend of the node size has been added to Figures 2D and 3D. We apologize once again for this mistake.

  1. Figure 4D, GO annotations are generally grouped into three main categories: Biological Process (BP), Cellular Component (CC) and Molecular Function (MF). Authors should show comprehensive GO enrichment results, or at least present them in the supplementary material.

Response: We have added the CC and MF result description to the manuscript (line 281-285). And the comprehensive GO enrichment results were also provided in Table S6.

  1. They carried metabolomics of cecal and hepatic, but only transcriptome of hepatic. Why? They performed metabolomics of the cecal and hepatic, but only the transcriptome of the hepatic. Please clarify.

Response: Studies have shown that the gut and liver communicate with each other through the portal vein, biliary tract, and systemic circulation. Metabolites produced by the microbiota in the gut can be transported to the liver and influence liver function by regulating gene expression in the liver.

In this study, we assumed that the gut-produced metabolites could be transported to the liver and influence hepatic lipogenesis by regulating gene expression in the liver. We first performed a metabolomic analysis of the cecum and liver to identify metabolites as potential signal molecules in the gut liver crosstalk (transported from the gut to the liver). Then we characterized the changes in hepatic gene expression profiles after HFD treatment and identified key genes associated with hepatic lipogenesis. Finally, we analyzed the correlation between the potential signal molecules and key genes, aiming to investigate the possible mechanisms that gut-liver regulate lipogenesis in chickens.

As mentioned above, we focus on the crosstalk between the gut and liver, especially the interplay between the gut-produced metabolite and the hepatic gene expression. Therefore, we detected both the metabolomics of cecal content and liver and only the hepatic transcriptomics.

  1. Same question. In section 3.5, the authors selected DAMs both in the cecum and liver with the same abundance trend between NFD and HFD chickens and then analyzed their correlation with the expression of DEGs, while the DEGs are based on the liver transcriptome. Could the author please explain it?

Response: We first selected DAMs both in the cecum and liver with the same abundance trend between NFD and HFD chickens. Because we want to identify metabolites as potential signal molecules in the gut liver crosstalk (transported from the gut to the liver), and DAMs both in the cecum and liver with the same abundance trend between NFD and HFD chickens are most likely as signal molecules between gut and liver. Then we identified DEGs based on the liver transcriptomic analysis because we wanted to identify potential genes regulated by the gut-produced metabolites.

  1. Line 307, "Ten DAMs overlayed between the two comparisons." But in Figure 5A, it is eleven.

Response: We are so sorry for this mistake. It should be eleven DAMs overlayed between the two comparisons, and ten DAMs showed a consistent abundance trend in the cecum and liver. We have corrected this mistake (line 359-361).

  1. Line 398, omit the word "genes".

Response: It has been deleted.

Thanks again for your suggestions!

Reviewer 3 Report

General comment#

In the study “Integrated metabolomic and transcriptomic analysis reveals potential gut-liver crosstalks in the lipogenesis of chicken”, Chen et.al investigated the gut-liver crosstalk using metabolomic and RNAseq methods and their role in lipogenesis in chicken models.

They established obese chicken model by inducing lipogenesis and fat deposition using high-fat diet (HFD). In HFD chickens, a significant difference in the metabolic profiles of the cecum and liver was detected using ultra high-performance liquid chromatography-tandem mass spectrometry (UHPLC-MS/MS) analysis. A total of 113 and 73 differentially abundant metabolites (DAMs) were detected in cecum and liver followed by a pathway enrichment analysis. A hepatic transcriptome analysis of chickens fed with NFD and HFD was performed identifying 271 DEGs (differentially expressed genes) (NFD vs HFD). Further investigation revealed potential role of detected metabolites (10 metabolites) in gut-liver crosstalks. At gene level profiling, 35 DEGs corresponding to 19 GO terms involved in lipid metabolic process was used for correlation analysis (with 10 common metabolites in liver) to identify metabolite-gene interactions between gut and liver, revealing two clusters. Study suggest gut to liver transport of 5 metabolites followed by up- and down-regulation of several genes in chicken liver which might influence the lipogenesis.

The study is well designed and the manuscript is well articulated. Enjoyed reading the article.

 Minor comments#

Line 87 - Change ‘twenty’ to ‘Twenty’

Line 131 – Capitalize ‘h’

Please explain about the violine plot given in Figure 1. A detailed legend would benefit the reader.

Lines 250-252 – Authors are mistaken here. Change to “The abundance of 2-Oxo-4-methylthiobutanoic acid, 4- Hydroxycinnamic acid, and ethyl oleate decreased both in the cecum and liver of chickens fed with HFD compared with chickens fed with NFD.”

Line 403 – Spelling mistake. Change ‘gur’ to ‘gut’

Please proofread the manuscript thoroughly for English including spelling check.

Author Response

Dear reviewer,

We really appreciate your helpful comments and valuable suggestions. The comments have been addressed and corrected. We hope that the present manuscript now is suitable for publication in Animals. All the corrections and modifications are as follows:

Reviewer #3

  1. Line 87 - Change 'twenty' to 'Twenty'

Response: It has been changed.

  1. Line 131 – Capitalize 'h'

Response: The “h” has been capitalized.

  1. Please explain about the violine plot given in Figure 1. A detailed legend would benefit the reader.

Response: A detailed legend has been added to Figure 1 (line 211-214).

  1. Lines 250-252 – Authors are mistaken here. Change to "The abundance of 2-Oxo-4-methylthiobutanoic acid, 4- Hydroxycinnamic acid, and ethyl oleate decreased both in the cecum and liver of chickens fed with HFD compared with chickens fed with NFD."

Response: It has been corrected.

  1. Line 403 – Spelling mistake. Change 'gur' to 'gut'

Response: “gur” has been changed to “gut”.

  1. Comments on the Quality of English Language

Please proofread the manuscript thoroughly for English including spelling check.

Response: We have proofread the manuscript for English and checked for spelling errors.

Thanks again for your suggestions!

Round 2

Reviewer 2 Report

I am generally satisfied with the author's response, and although I still think it does not make complete sense that only the transcriptome of the hepatic was done, the author's explanation seems to be reasonable.  I would suggest that the authors note this issue after future work, especially since transcriptomes are not that expensive. I think this paper is appropriate for publication in Animals.